# Abnormal Circulating Maternal miRNA Expression Is Associated with a Low (<4%) Cell-Free DNA Fetal Fraction

**DOI:** 10.3390/diagnostics11112108

**Published:** 2021-11-14

**Authors:** Graziano Santoro, Cristina Lapucci, Marco Giannoccaro, Simona Caporilli, Martina Rusin, Anna Seidenari, Maurizio Ferrari, Antonio Farina

**Affiliations:** 1Genetic Unit, Synlab, Via B. L. Pavoni 18, Castenedolo, 25014 Brescia, Italy; graziano.santoro@synlab.it (G.S.); cristina.lapucci@synlab.it (C.L.); marco.giannoccaro@synlab.it (M.G.); simona.caporilli@agilent.com (S.C.); 2Division of Obstetrics and Prenatal Medicine, Department of Medicine and Surgery (DIMEC), IRCCS Sant’Orsola-Malpighi Hospital, University of Bologna, 40138 Bologna, Italy; rusin.martina@gmail.com (M.R.); annaseidenari@gmail.com (A.S.); antonio.farina@unibo.it (A.F.); 3IRCCS, SDN, Via Gianturco 113, 80143 Naples, Italy

**Keywords:** miRNA, NIPT, low cell-free DNA (cfDNA) fetal fraction

## Abstract

The present pilot study investigates whether an abnormal miRNA profile in NIPT plasma samples can explain the finding of a low cell-free DNA (cfDNA) fetal fraction (cfDNAff) in euploid fetuses and non-obese women. Twelve women who underwent neoBona^®^ NIPT with a normal fetal karyotype were studied. Six with a cfDNAff < 4% were matched with a control group with normal levels of cfDNAff > 4%. Samples were processed using the nanostring nCounter^®^ platform with a panel of 800 miRNAs. Four of the maternal miRNAs, miR-579, miR-612, miR-3144 and miR-6721, had a significant abnormal expression in patients. A data filtering analysis showed that miR-579, miR-612, miR-3144 and miR-6721 targeted 169, 1, 48 and 136 placenta-specific genes, respectively. miR-579, miR-3144 and miR-6721 shared placenta-specific targeted genes involved in trophoblast invasion and migration pathways (IGF2R, PTCD2, SATB2, PLAC8). Moreover, the miRNA target genes encoded proteins localized in the placenta and involved in the pathogenesis of pre-eclampsia, including chorion-specific transcription factor GCMa, PRG2, Lin-28 Homolog B and IGFBP1. In conclusion, aberrant maternal miRNA expression in circulating plasma could be a source of dysregulating trophoblast invasion and migration and could represent a novel cause of a low cfDNAff in the sera of pregnant women at the time of NIPT analysis.

## 1. Introduction

Non-invasive prenatal testing (NIPT) is a sophisticated method used worldwide for testing chromosomal abnormalities in the fetus. Currently, NIPT is considered to be a valuable test for pregnant women to provide a cost-effective and highly efficient early screening of genetic disorders. The high detection rate of trisomy and the increased choice of NIPT methods available on the market increased the number of pregnant women worldwide (estimated 4–6 million per year) who undergo the test annually [1]. The NIPT method is based on the detection of the cell-free DNA (cfDNA) fetal fraction (cfDNAff), in which the concentration can vary in the maternal blood from 4 to 30% [2]. Because the fetal component of the circulating DNA comes from the placenta, the NIPT screening test is still considered to have discordant results compared to the fetal karyotype. This is due to maternal chromosomal rearrangements or mosaicism, maternal malignancy, confined placental mosaicism, or vanishing twin pregnancies [3,4,5]. The testing outcome of NIPT depends on the cfDNAff concentration detected in the blood sample; each NIPT method has a minimum cfDNAff concentration threshold (usually 4%) below which the test results are invalid. The reasons for a low cfDNAff are still under investigation but they include a higher body mass index (BMI) [6], fetal aneuploidies [6], anticoagulation therapy [7] and, perhaps, fetal growth restriction (FGR) and pre-eclampsia (PE) [8,9]. More recently, other cell-free nucleic acids, such as fetal DNA in maternal urine, mRNA, and microRNAs (miRNAs), have become promising targets for prenatal diagnostic applications [10,11]. Extensive research has been carried out in recent years on circulating miRNAs and mRNA, and their role as potential clinical biomarkers for the prediction of pregnancy complications [12], such as PE, congenital heart diseases and gestational diabetes (GDM) [11,13,14,15]. MicroRNAs are well known for their role in post-transcriptional gene silencing in a sequence-specific manner by means of recognizing 3′UTR or 5′UTR regions of target mRNAs [16]. It was estimated that in humans, miRNAs modulate up to 60% of protein-coding genes in a ubiquitous manner. For this reason, they are involved in several biological processes, such as differentiation, proliferation, apoptosis and development [17]. The fine-scale adjustment role of miRNAs in protein output leads, in cases of dysregulation, to human diseases, such as diverse types of cancer, several types of neurological disorders [18] and, as previously described, pregnancy complications [12,19].

This preliminary study analyzed the miRNA expression profile of two groups of women stratified according to cfDNAff (Group A < 4% and Group B > 4%) at 10–14 weeks gestation to find a possible genetic reason capable of explaining a low cfDNAff < 4%.

## 2. Materials and Methods

### 2.1. Study Design

A retrospective cohort study was carried out involving a 1:1 match of non-consecutive patients between January 2016 and June 2019, including 12 pregnant women who had undergone NIPT from 10 + 0 to 14 + 1 gestational weeks as a primary or secondary screening test for fetal chromosomal abnormalities. All patients gave their informed consent for clinical and research purposes including NIPT and miRNAs analyses. The study was conducted in accordance with the ethical standards for human research of the Declaration of Helsinki. The cohorts were generated according to cfDNAff values.

### 2.2. Samples Selection

The sample selection was made based on the NIPT results (neoBona^®^ test). Blood samples were collected from Caucasian pregnant women, put into cfDNA BCT^®^s (Streck, La Vista, NE, USA).

Twelve plasma samples, already isolated and analyzed for NIPT test, were selected and divided into 2 groups: 6 samples with a normal cfDNAff > 4% (group B) and 6 samples with a cfDNAff < 4% (group A). The sample selection within each group was matched for BMI, gestational age (GA) and maternal age.

### 2.3. Total RNA Extraction from Plasma

The total RNA was isolated from 2 mL of plasma using the QIAamp Circulating Nucleic Acid Extraction Kit (Qiagen, NE) according to the manufacturer’s instructions. The lysis procedures specified in the RNA extraction protocol were followed by adding 15 µL of synthetic control RNA spike-in oligo mixture (previously aliquoted and stored at −20 °C at 1–5 pg/μL) to each sample in order to check for variances in the starting material as well as for the efficiency of the downstream total RNA extraction step, according to NanoString’s instructions (Nanostring Technologies, Seattle, WA, USA). RNA samples were treated with DNAse I (Life Technologies, Carlsbad, CA, USA) and purified by using the RNeasy MinElute Cleanup kit (Qiagen, NE) followed by Amicon ultra 0.5 mL centrifugal filters (Merck KGaA, GE) in the high nucleic acid concentration step according to the manufacturer’s instructions. The RNA samples were assessed for quantity control using the Qubit^®^ RNA HS (High Sensitivity) assay kit of the Qubit 2.0 Fluorometer (Life Technologies, USA).

### 2.4. NanoString nCounter Human v3 miRNA Assay

Sample preparation for the nCounter miRNA analysis was carried out using the NanoString Human v3 miRNA assay (Nanostring Technologies, USA) according to the manufacturer’s instruction. For each sample, a scan of 550 of the field of view (FOV) was imaged. A sample run quality control check was carried out before data normalization by processing the nCounter data imaging QC metrics which showed no discrepancy between the FOVs attempted and those counted, and the binding density for the samples which was within the recommended range. The reporter code count (RCC) file produced by the nCounter^®^ Digital Analyzer was loaded into the nSolverTM Analysis Software 4.0 (Nanostring Technologies, USA). Data normalization involved the geometric mean of the positive controls for code count normalization, nCounter probed background adjustment with a threshold set to 30 (which is a value both bigger and smaller than the mean values of the negative and positive controls, respectively) and the normalization of the sample input amount to the geometric means of the housekeeping endogenous miRNA controls β-actin (ACTB), Beta-2-Microglobulin (B2M), Glyceraldehyde-3-Phosphate Dehydrogenase (GAPDH), Ribosomal Protein L19 (RPL19) and Ribosomal Protein Lateral Stalk Subunit P0 (RPLP0) in both the test and the control group samples.

### 2.5. Data Analysis

For the samples of both groups, the miRNAs expression raw data were transformed into fold-of-change using a linear scale, which was calculated by comparing the code count of each sample in test Group A to the geometric mean in control Group B (the expression value of which was normalized to 1). The result outcomes showed relative expression defects with negative and positive values indicating under- and over-expression versus control Group B respectively. miRNA expression of 1 in the Group A samples meant no differences to control Group B. The values extracted between the two groups were statistically significant with a *p*-value < 0.05. The miRNAs with expression defects of at least ±1.5-fold change were eligible for a final selection during data filtering. A bioinformatic analysis was carried out on the miRNAs, respecting the fold-of-change criteria. Target scan (targetscan.org/vert_72/; accessed on 30 March 2020) webserver was used to predict the target genes of the miRNAs selected, which were launched into the Panther Classification system (http://pantherdb.org; accessed on 30 March 2020) to predict the pathway distribution within annotated biological processes. The data filtering of the target genes resulted in a group of genes shared among the miRNAs selected. Each gene was launched into the Uniprot webserver (uniprot.org; accessed on 30 March 2020) to predict molecular functions, biological processes, cell localization and annotated diseases. A human protein atlas was used to select tissue-specific enriched genes on the list of miRNA target genes selected (proteinatlas.org; accessed on 31 March 2020).

## 3. Results

### 3.1. Demographic Data

Table 1 shows the distribution of the clinical variables available expressed as the median (min-max) of the two groups. No differences were found for BMI and gestational age, the two major known factors influencing the cfDNAff values. No fetal aneuploidies were present in the study groups. No fetal or maternal diseases were recorded, including pre-eclampsia, gestational diabetes mellitus and preterm delivery; however, in Group A, the average neonatal weight was lower due to the presence of two FGR cases detected at 30–32 weeks gestation during a routine ultrasound scan.

### 3.2. Evaluation of Placenta-Specific miRNAs with Changes of Expression

Samples in both groups were tested using a miRNA panel provided by NanoString Technologies, Inc. The nCounter^®^ Human v3 miRNA expression assay kit contains 800 miRNAs selected to be both clinically relevant and/or sequenced with high confidence. Within the panel of 800 miRNAs, 157 (20%) were placental-related. The group of placenta-related miRNAs included four categories involving placenta-specific miRNAs (expressed largely or uniquely in the placenta), placenta-associated miRNAs (ubiquitously expressed in the placenta and other tissues), circulating placenta-derived miRNAs (placenta-released circulating miRNAs) and uterine miRNAs (specifically expressed in the uterus) [20]. The miRNA expression values of the control group samples were normalized to 1, and the miRNA expression obtained in the test group samples was considered statistically significant at a *p*-value < 0.05.

Figure 1 shows the flow chart of miRNA data analysis and selection; 570 (71%) miRNAs in the Group A samples mirrored expression seen in the miRNAs of the control Group B samples, and 230 (29%) miRNAs showed variable expression signals. A significant number of placenta-related miRNAs also belonged to placenta-specific chromosomal clusters as reported in Table 2.

However, of the 10 downregulated miRNAs, some were previously reported to be associated with pregnancy disorders, such as PE, FGR and preterm delivery (PTD) [21]. For example, in accordance with Higashijima A et al. [21], the present study found a downregulation of miR-519 which had been associated with FGR. Moreover, the downregulation of miR-517-5p was also found in a study by Hromadnikova [22] to be associated with PE.

Ten placenta-related miRNAs showed under-expression in Group A when compared with Group B (Table 3) but, unfortunately, with a fold-change not exceeding the value −1.

### 3.3. Evaluation of miRNAs with Significant Changes of Expression of at Least ±1.5-Fold

Fifty-one maternal origin miRNAs, therefore non-placenta-associated, had a fold change in Group A within ±1. Four of the 51 miRNAs showed a greater expression of at least ±1.5-fold and satisfied the criteria of inclusion for additional analyses. Three of them, hsa-miR-612, hsa-miR-3144-3p and hsa-miR-6721-5p, were downregulated. The only upregulated miRNA was hsa-miR-6721-5p.

### 3.4. Predicted Biological Processes for the Four miRNA Target Genes Selected

The Targetscan v7.0 platform was used to identify the validated targets of the four miRNAs of interest. The potential target genes for miR-579, miR-612, miR-3144 and miR-6721 were 7013, 31, 2333 and 6238, respectively. Each targeted gene list was then launched into the Panther Classification system platform to obtain a distribution within the annotated biological processes. A comparison among the lists of the miRNA targeted genes selected showed that, together, they shared eight biological processes. The miR-579 targeted genes were the most represented in each biological process identified. In contrast, the miR-612 targeted genes were the least represented in all processes (data not shown).

### 3.5. Selected Maternal miRNAs and Placenta-Specific Genes

It was investigated if any of the four selected maternal miRNAs were involved in the expression of placenta-specific genes; 494 annotated placenta-specific genes were available on the human protein atlas (proteinatlas.org; accessed on 31 March 2020). Interestingly, a data filtering analysis of selected miRNAs showed that miR-579, miR-612, miR-3144 and miR-6721 target 169, 1, 48 and 136 placenta-specific genes, respectively. Using a Venn diagram, as shown in Figure 2, it was observed that miR-579, miR-3144 and miR-6721 shared eight placenta-specific targeted genes. Since miR-612 did not share any gene, it was excluded from the Venn diagram. Each gene was launched into the Uniprot webserver and classified using Gene Ontology categories to better identify the biological processes in which these genes are involved (Table 4). The most annotated were Special AT-Rich Insulin like Growth Factor 2 Receptor (IGF2R), Pentatricopeptide Repeat Domain (PTCD2) and Sequence-Binding Protein 2 (SATB2). Of note, miR-579-3p and miR-3144-3p shared the Placenta Associated 8 (PLAC8) gene. The PLAC8 gene, the expression of which is enriched in the placenta more than in other tissues, promotes trophoblast invasion and migration [28]. Both IGF2R and SATB2 were also shown to be involved in the regulation of apoptotic process post-embryonic development and embryonic pattern specification.

### 3.6. Placental Proteins Encoded by miRNAs Target Genes

Finally, whether the maternal miRNA targeted genes selected encoded the proteins localized in the placental tissues was also investigated. It was found that miR-579 targeted (1) The glial cells missing transcription factor1 (GCM1) gene which encodes chorion-specific transcription factor GCMa localized in the syncytiotrophoblasts and associated with PE [29]; (2) The Proteoglycan 2 (PRG2) gene encoding Proteoglycan 2 localized in the extravillous trophoblasts [30]; (3) The lin-28 homolog B (LIN28B) gene coding for Lin-28 Homolog B which inhibits the invasion of trophoblast cells and the development of placental vasculature, and participates in the occurrence of PE [31]; (4) Insulin like growth factor binding protein 1 (IGFBP1) localized in the decidual cells. The GCM1 and PRG2 genes were also targeted by miR-6721; however, LIN28B was localized only by miR-3144.

## 4. Discussion

MicroRNA expression defects were studied in cancer as either oncogenes or tumor suppressors highlighting their potential role as non-invasive biomarkers for the diagnosis and prognosis of cancer [32]. The miR-579-3p gene was studied as an oncosuppressor of human melanoma [33] and, similarly, miR-612 was investigated as a tumor suppressor of several types of cancer [34,35]. This study analyzed the expression profile of circulating miRNAs from blood samples drawn from residual sera of pregnant women at 10 + 0 to 14 + 1 weeks of gestation who had undergone NIPT. The miRNA expression profile was investigated after a dichotomic sample selection based on the cfDNAff level detected by the NIPT test. It was hypothesized that a low cfDNAff is an epiphenomenon of insufficient trophoblast invasion, differentiation and development related to an abnormal panel of miRNA expression. As support for the present hypothesis, the authors found four maternal miRNAs with a significantly aberrant value of expression, and many others (including some placenta-specific miRNAs) with a less pronounced abnormal value, not however exceeding 1.5-fold. For the four maternal miRNAs with a more than a 1.5-fold-change, the most relevant target genes were IGF2R, PTCD2, SATB and PLAC8. They were all demonstrated to have direct trophoblast activity. The findings in the present study are in line with some previously described studies in which a possible maternal control of placental development, fetal growth and infant outcomes via miRNAs were suggested [36,37]. To the authors’ knowledge, it was, however, the first time that an abnormal maternal miRNA expression was associated with a low cfDNAff due to possible trophoblast dysregulation.

The IGF2R interacts with insulin-like growth factor 2 (IGF2) to regulate several biological processes including cell growth, survival and migration [38]. Evidence has shown that IGF2R enhances proliferation and apoptosis of human first-trimester cytotrophoblasts which is the source of cffDNA by trophoblast proliferation and survival [39]. It is possible that the dysregulation of IGF2R signaling produces an abnormal cytotrophoblast turnover and, consequently, a lower cfDNAff.

The PTCD2 gene belongs to a large family of RNA-binding proteins known as Pentatricopeptide repeat (PPR) proteins; PTCD2 expression was found with the highest levels in the heart, liver and kidneys [40]. However, according to the human protein atlas, this gene is also highly expressed in the trophoblast cells.

The SATBs are well-known as regulators of chromatin organization and are involved in several processes, such as apoptosis, cell invasion, metastasis, proliferation, angiogenesis and immune modulation. Within the SATB family, both SATB-1 and SATB-2 were investigated in cancer progression [41] and are involved in human placenta development by promoting trophoblast stem cell renewal and inhibiting their differentiation [42]. The expression of the PLAC8 gene (targeted only by miR-579 and miR-3144) is enriched in the placenta more than in other tissues, and it is specifically expressed in the interstitial extravillous trophoblast cells on the fetomaternal interface to promote trophoblast invasion and migration [28]. These findings suggested that the miRNA targeted genes selected play an important role during proper placental development, and it is plausible that reduced trophoblast activity results in a lower circulating cfDNAff.

The tissue-specificity of the miRNA target genes selected was therefore investigated, and it was found that a significant number were expressed in the placenta more than in any other tissues; miR-579 showed the highest number of placenta-related target genes, eight of which shared placenta-related target genes with the other miRNAs (Figure 2).

Finally, the authors also found that the miRNA target genes selected encoded proteins, including GCM1 PRG2 LIN28B and GFBP1 typical of the placental tissue, and were also involved in PE, a disease in which the cfDNAff is lower than expected [9,43]. It should also be noted that, in Group A, there were two cases of FGR, another putative condition that seems to be associated with a lower cfDNAff [8].

## 5. Conclusions

In this study, it was hypothesized that the dysregulation of not-well understood molecular pathways affecting trophoblast invasion and migration, as a consequence of the altered expression of a panel of maternal miRNAs at the time of NIPT, was a cause of a low cfDNAff.

This preliminary study shows very promising data. If confirmed in larger cohort studies, the miRNA panel identified in this study could be used as a biomarker to understand some of the maternal-fetal interactions which lead to a low cfDNAff at the time of routine NIPT.

## Figures and Tables

**Figure 1 diagnostics-11-02108-f001:**
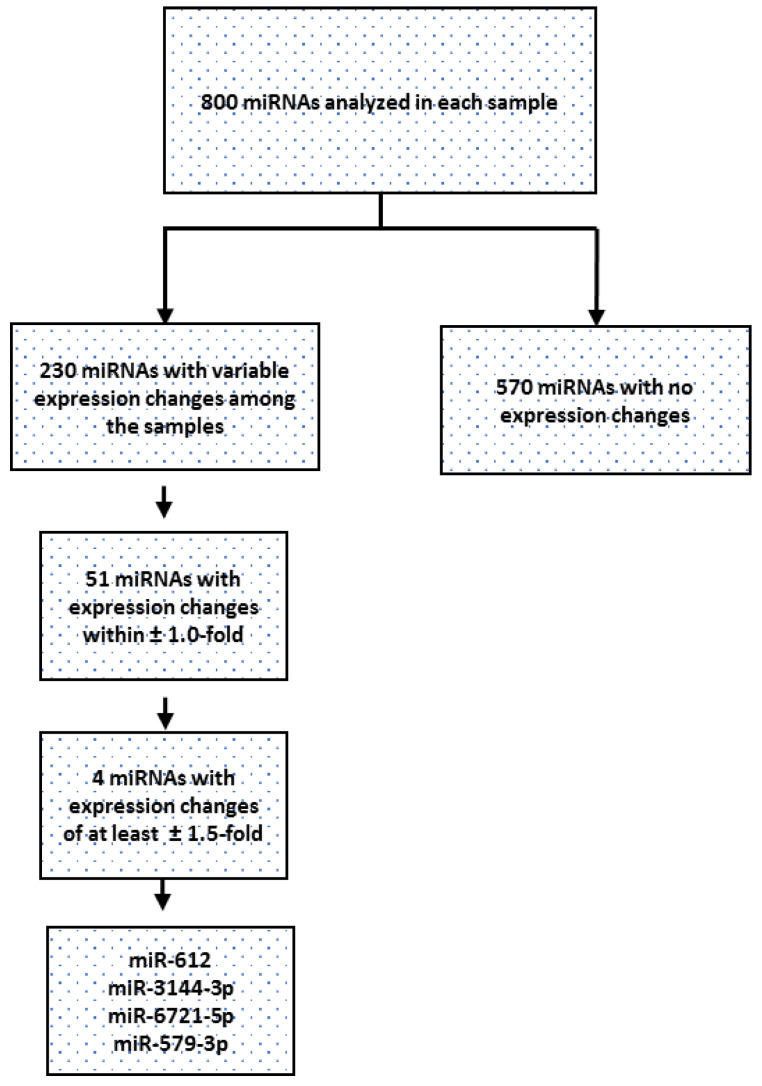
Schematic representation of the miRNA data analysis. The miRNAs data filtering in each sample of the test group was set on the basis of the significant expression defects compared to the miRNAs in the control group samples. Expression changes of the 4 miRNAs selected (miR-612, miR-3144, miR-6721 and miR-579) were ± ≥1.5-fold.

**Figure 2 diagnostics-11-02108-f002:**
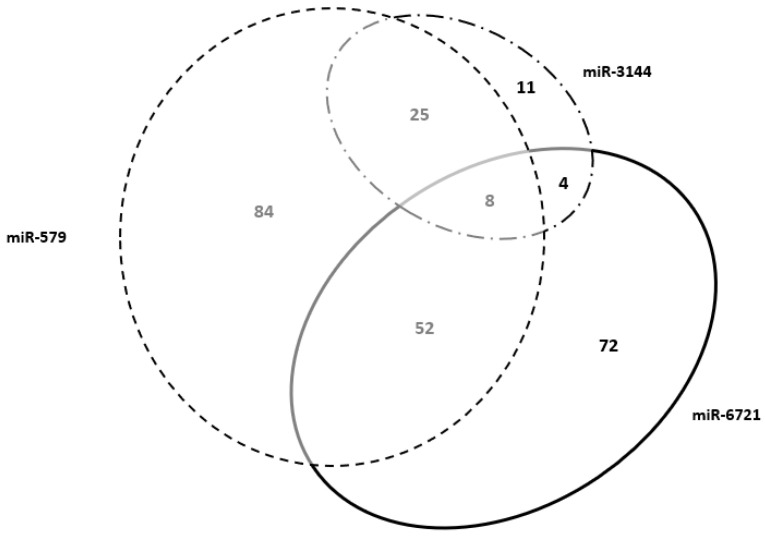
miRNA placenta-specific target genes. A Venn diagram illustrating that the highest number of placenta-specific target genes are related to miR-579 (169) and to miR-6721 (136) which share 52 genes (31% and 38% over the total number of placenta-specific genes, respectively). miR-3144 targeted 48 placenta genes of which 25 (52%) are shared with miR-579 targeted placenta-specific genes. A list of miRNA placenta-specific target genes is reported in Table 4.

**Table 1 diagnostics-11-02108-t001:** Clinical characteristics of the samples. Data are expressed as median (min-max) or percentage.

Variable	Group A cfDNAff < 4% (*n* = 6)	Group B cfDNAff > 4% (*n* = 6)	*p*-Value *
GA (days)	82.0 (72–99)	84.5 (71–91)	>0.99
BMI	22.48 (20.43–27.72)	20.03 (17.26–25.82)	0.065
Maternal age (yrs)	33.7 (29–42)	33.6 (26–41)	>0.99
Female (%)	50	33.3	0.500
cfDNAff	0.5 (0–1)	15.5 (15–17)	-
Neonatal weight (gr.)	2955 (2100–3140)	3080 (2860–3680)	0.121
FGR (%)	33.3	0	<0.001

* Mann-Whitney or Fisher exact test Abbreviations: GA: gestational age; BMI: body mass index; cfDNAff: cell-free DNA fetal fraction; FGR: fetal growth restriction.

**Table 2 diagnostics-11-02108-t002:** Placenta-related miRNAs (microRNAs) and their placenta-specific chromosomal clusters.

miRNAs	Localization
C19MC14 [23]	chromosome 19 encoding 46 intronic miRNAs
C14MC11 [19]	on chromosome 14 encoding 39 miRNAs
placenta-specific 1302 miRNA family [24]	derived from the MER53 transposon element
as miR-371/miR-373 [25]	chromosome 19 encoding 7 miRNAs
miR-17-92 11. [26]	chromosome 13 encoding for 6 miRNAs
miRNA 148-152 family [27]	chromosomes 7, 12 and 17 encoding 3 miRNAs

**Table 3 diagnostics-11-02108-t003:** Placenta-related miRNAs showing aberrant negative values (within −1 fold) in Group A with a cfDNAff < 4%.

Placenta-Related miRNAs
hsa-let-7a-5p
hsa-miR-210-3p
hsa-miR-369-3p
hsa-miR-371a-5p
hsa-miR-512-5p
hsa-miR-517a-3p
hsa-miR-519a-3p
hsa-miR-93-5p
hsa-miR-96-5p
hsa-miR-206

**Table 4 diagnostics-11-02108-t004:** List of miRNA placenta-specific target genes of the three miRNAs as represented in Venn diagram (Figure 2) with Gene Ontology (GO) categories.

miRNAs	Placenta-Specific Target Genes	Gene Ontology
miR-579	*ADAMTSL4*, *ADM*, *AFF1*, *AFF2*, *AGTR1*, *AKR1B15*, *ARL14EPL*, *ATP6V1C2*, *BCAR3*, *C18orf54*, *C2orf83*, *CCDC102B*, *CLEC1A*, *COBLL1*, *COL11A1*, *CRYBG1*, *CYP19A1*, *DAB2*, *DKK1*, *EDNRB*, *EPB42*, *EPYC*, *ERVV-1*, *ESRRG*, *F13A1*, *F5*, *FAM162B*, *FAM46A*, *FBN1*, *FBN2*, *GDF6*, *GJB7*, *GM2A*, *GPR1*, *HPGDS*, *HSD17B2*, *HTR1F*, *HTR2B*, *IDO2*, *IGFBP1*, *IGSF3*, *IL1RAP*, *IL1RL1*, *JAM2*, *KATNBL1*, *LUM*, *LYPD6*, *MAGEA10*, *MB21D2*, *MEST*, *MFAP5*, *MUC15*, *NLRP10*, *NNAT*, *NRK*, *OLIG3*, *OLR1*, *PAPPA*, *PEG3*, *PHLDA2*, *PHLDB2*, *PKIB*, *PSG1*, *PSG2*, *PSG4*, *PSG8*, *RASA1*, *SEMA6D*, *SIGLEC6*, *SLC25A35*, *SPP1*, *STS*, *TBX4*, *TBX5*, *TFAP2A*, *TFRC*, *TMEM2*, *TWIST1*, *TXK*, *VGLL1*, *VGLL3*, *WNT2*, *XKRX*, *ZNF468*	cellular process (GO:0009987)
		metabolic process (GO:0008152)
		biological regulation (GO:0065007)
		localization (GO:0051179)
		multicellular organismal process (GO:0032501)
		response to stimulus (GO:0050896)
		biological adhesion (GO:0022610)
		developmental process (GO:0032502)
		immune system process (GO:0002376)
		cellular component organization or biogenesis (GO:0071840)
		reproduction (GO:0000003)
		cell proliferation (GO:0008283)
		rhythmic process (GO:0048511)
		biological phase (GO:0044848)
		nitrogen utilization (GO:0019740)
miR-3144	*C2orf72*, *ELOVL2*, *HAPLN1*, *MORN3*, *P2RY1*, *SKP2*, *SLC30A2*, *TFPI2*, *TRAPPC3L*, *TRIM10*, *TRIM64B*	cellular process (GO:0009987)
		metabolic process (GO:0008152)
		biological regulation (GO:0065007)
		localization (GO:0051179)
		multicellular organismal process (GO:0032501)
		response to stimulus (GO:0050896)
		developmental process (GO:0032502)
		biological adhesion (GO:0022610)
		locomotion (GO:0040011)
		pigmentation (GO:0043473)
miR-6721	*AADACL3*, *ALPP*, *AOC1*, *APLNR*, *ARID3A*, *ASCL2*, *ATG9B*, *BIRC7*, *C1QTNF6*, *CAPN6*, *CD248*, *CDH5*, *CLDN19*, *CLDN9*, *COL4A2*, *COX4I2*, *CXorf67*, *CYP11A1*, *DACT2*, *DLX3*, *DUSP9*, *ERVFRD-1*, *ERVV-2*, *FAM129B*, *FSTL3*, *FURIN*, *GABRE*, *GDPD5*, *GJA5*, *GPR78*, *GRHL2*, *HES2*, *INSL4*, *ISM2*, *KLF14*, *LAMC3*, *LARGE2*, *LEP*, *LGR5*, *MAFK*, *MFSD2B*, *N4BP3*, *NOTUM*, *NOX5*, *OC90*, *PGF*, *PODNL1*, *PROCR*, *SDC1*, *SEMA7A*, *SH2D5*, *SLC13A4*, *SLC22A11*, *SLC43A2*, *SLC4A1*, *SLC7A4*, *SP6*, *ST3GAL4*, *STRA6*, *SYNPO*, *SYTL5*, *TIMP2*, *TMEM139*, *TNS4*, *TREML2*, *TRIM58*, *TRPV6*, *UNC13D*, *WNT1*, *WNT3A*, *WNT7A*, *ZFAT*	cellular process (GO:0009987)
		metabolic process (GO:0008152)
		biological regulation (GO:0065007)
		localization (GO:0051179)
		multicellular organismal process (GO:0032501)
		response to stimulus (GO:0050896)
		developmental process (GO:0032502)
		immune system process (GO:0002376)
		biological adhesion (GO:0022610)
		cellular component organization or biogenesis (GO:0071840)
		reproduction (GO:0000003)
		cell proliferation (GO:0008283)
		rhythmic process (GO:0048511)
		biological phase (GO:0044848)
		signaling (GO:0023052)
		pigmentation (GO:0043473)
		multi-organism process (GO:0051704)
Genes shared between miR-579 and miR-6721	*ACKR2*, *ADAM12*, *ADGRG6*, *APLN*, *ART4*, *CADM3*, *CCSAP*, *CD59*, *CLDN6*, *CREB3L2*, *CYTH3*, *DLK1*, *EGFL7*, *EXPH5*, *FHDC1*, *GADD45G*, *GCM1*, *GDF15*, *GRAMD2A*, *GSE1*, *HPGD*, *IGDCC3*, *IGF2*, *IGF2BP1*, *IL2RB*, *INHBA*, *ITIH5*, *KMO*, *LRRC15*, *LVRN*, *MBNL3*, *MORC4*, *NCMAP*, *PAEP*, *PDPN*, *PLAU*, *PRG2*, *QSOX1*, *RAI14*, *RGPD1*, *RSPO2*, *RXFP1*, *SCIN*, *SERPINE1*, *SLC2A1*, *SLC38A9*, *SLC6A2*, *TGM2*, *THSD7A*, *TIMP3*, *UCK2*, *ZFP42*	cellular process (GO:0009987)
		metabolic process (GO:0008152)
		biological regulation (GO:0065007)
		localization (GO:0051179)
		response to stimulus (GO:0050896)
		developmental process (GO:0032502)
		immune system process (GO:0002376)
		biological adhesion (GO:0022610)
		cell proliferation (GO:0008283)
Genes shared between miR-579 and miR-3144	*ADAMTS20*, *ADAMTS5*, *ATP10D*, *CHSY1*, *CYSLTR2*, *DEPDC1B*, *FN1*, *GULP1*, *HGF*, *HMGB3*, *LGALS13*, *LIN28B*, *LIPG*, *LNPEP*, *MAN1A2*, *NECTIN3*, *NFE2L3*, *NIPAL1*, *PLAC8*, *SH3TC2*, *SPTLC3*, *TDRP*, *TFPI*, *TLR3*, *TUSC3*	cellular process (GO:0009987)
		metabolic process (GO:0008152)
		biological regulation (GO:0065007)
		localization (GO:0051179)
		multicellular organismal process (GO:0032501)
		response to stimulus (GO:0050896)
		developmental process (GO:0032502)
		biological adhesion (GO:0022610)
		immune system process (GO:0002376)
		reproduction (GO:0000003)
		cellular component organization or biogenesis (GO:0071840)
Genes shared between miR-3144 and miR-6721	*ERVMER34-1*, *FAM89A*, *MFAP2*, *SPIRE2*	cellular process (GO:0009987)
		metabolic process (GO:0008152)
		biological regulation (GO:0065007)
		localization (GO:0051179)
		multicellular organismal process (GO:0032501)
		response to stimulus (GO:0050896)
		developmental process (GO:0032502)
		biological adhesion (GO:0022610)
Genes shared Among miR-579, miR-3144 and miR-6721	*ARHGAP42*, *FOXO4*, *GUCY1A2*, *IGF2R*, *PHACTR2*, *PTCD2*, *SATB2*, *VAV3*	cellular process (GO:0009987)
		metabolic process (GO:0008152)
		biological regulation (GO:0065007)
		localization (GO:0051179)
		multicellular organismal process (GO:0032501)
		response to stimulus (GO:0050896)
		developmental process (GO:0032502)
		biological adhesion (GO:0022610)

## Data Availability

Raw data were generated in the laboratory of Synlab Italy (Castenedolo, BS, Italy). The data derived supporting the findings of this study are available from the corresponding author upon request.

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
