# Peer review of "Abnormal Circulating Maternal miRNA Expression Is Associated with a Low (<4%) Cell-Free DNA Fetal Fraction"

_diagnostics, 2021, doi:10.3390/diagnostics11112108_

Round 1

Reviewer 1 Report

Summary: The manuscript investigates NPT analysis of 12 pregnant women from 10+0 to 14+1 gestational weeks to screen for fetal chromosomal abnormalities. From the analysis, 4 miRNAs were selected whose expression changes exceed 1.5 folds.

Strengths: The results show placenta-specific target genes that are related to miR-579 are  169, miR-6721 are 136, and miR-3144 are 48. Common gene targets of the miRs are also shown. 

Several analysis and computational techniques have been used to obtain new information.

Some genes are mentioned AT-Rich Insulin Like Growth Factor 2 Receptor (IGF2R), Pentatricopeptide Repeat Domain (PTCD2), Sequence-Binding Protein 2 (SATB2), Placenta Associated 8 (PLAC8), GCM1, PRG2, LIN28B, IGFBP1.

Weaknesses:

What is the basis of mentioning only the above genes among all the genes that were identified?

Lists should be shown for the genes which are represented in figure 2 - Venn diagram. 

Some kind of basic pathway analysis of the gene lists needs to be added to the study to highlight the biological significance of the observations.

Full-form of NIPT needs to be added in the abstract

Section 0 needs to be removed

Recent relevant references are missing e.g.

https://bjgp.org/content/67/660/298

https://bmcmedgenomics.biomedcentral.com/articles/10.1186/s12920-021-00941-y

https://bmjopen.bmj.com/content/6/1/e010002 

Author Response

1-What is the basis of mentioning only the above genes among all the genes that were identified?

Lists should be shown for the genes which are represented in figure 2 - Venn diagram. 

In our idea the 4 genes well described are the only ones that have a statistic expression variation and a correlation with pregnancies. otherwise as you suggested we added a table (table 4) with genes list represented in figure 2. 

2- Some kind of basic pathway analysis of the gene lists needs to be added to the study to highlight the biological significance of the observations.

We add in the table 4 the Gene Ontology categories to describe the biological processes in which target genes are involved. In paragraph 4 (Discussion) we’d elaborated on previously the pathways of the 4 genes we selected. 

If the table is too long we can add it in a supplementary data.

3- Full-form of NIPT needs to be added in the abstract

Point taken

4- Section 0 needs to be removed

Point taken

5- Recent relevant references are missing e.g.

https://bjgp.org/content/67/660/298

https://bmcmedgenomics.biomedcentral.com/articles/10.1186/s12920-021-00941-y

https://bmjopen.bmj.com/content/6/1/e010002

Point taken. 

Reviewer 2 Report

In this paper the authors analyzed the miRNA expression prfileof 2 groups of women stratified according to cfDNAff at 10-14 weeks gestation to find a possible genetic reason capable of explaining a low cfDNAff <4%.

Major revisions

Although the study is valid from a conceptual point, the samples analyzed are too few to consider a relationship between miRNA and low fetal fraction. The study showed by the authors seems a proof of concept that should be confirmed in a larger cohort.

Minor revisions

-paragraph 0 shol be removed

  • Among the possible causes of low fetal fraction, the authour should mention anticoaugulation therapy.
  • lane 41, I suggest to remove  early diagnosis and to replace with early screening
  • authors contribution and infromed consent are incomplete
  • lane 47, the cut off is ≡4% and not <4%

Author Response

Major revisions
1- Although the study is valid from a conceptual point, the samples analyzed are too few to consider a relationship between miRNA and low fetal fraction. The study showed by the authors seems a proof of concept that should be confirmed in a larger cohort.

In the conclusion paragraph we underlined the limit of the study: this is a pilot study, a larger cohort is needed to validate the hypothesis that the three miRNAs we found are involved in FF assessment. However, we stressed on this theme adding it in the abstract, in the introduction and remarked it in the conclusion.

Minor revisions
2-paragraph 0 shol be removed
Point taken
3-Among the possible causes of low fetal fraction, the authour should mention anticoaugulation therapy. 
Point taken

4-lane 41, I suggest to remove  early diagnosis and to replace with early screening
Point taken.
5-authors contribution and infromed consent are incomplete
Point taken.
6-lane 47, the cut off is ≡4% and not <4%
Point taken.

Round 2

Reviewer 1 Report

In the revised version, authors have addressed the concerns.

Manuscript is improved, may now be accepted.

Reviewer 2 Report

In this paper the authors analyzed the miRNA expression prfileof 2 groups of women stratified according to cfDNAff at 10-14 weeks gestation to find a possible genetic reason capable of explaining a low cfDNAff <4%.

The authors replied to all my comments.